# Effective Blue Light-Absorbing AuAg Nanoparticles in InP Quantum Dots-Based Color Conversion

**DOI:** 10.3390/ma15238455

**Published:** 2022-11-27

**Authors:** Hyo-Jin Yeo, Suk-Young Yoon, Dae-Yeon Jo, Hyun-Min Kim, Jeonghun Kwak, Sung-Phil Kim, Myung-Joon Kim, Heesun Yang

**Affiliations:** 1Department of Materials Science and Engineering, Hongik University, Seoul 04066, Republic of Korea; 2Department of Electrical and Computer Engineering, Seoul National University, Seoul 08826, Republic of Korea; 3Department of Biomedical Engineering, Ulsan National Institute of Science and Technology, Ulsan 44919, Republic of Korea; 4Department of Big Data Application, Hannam University, Daejeon 34430, Republic of Korea

**Keywords:** color-by-blue, AuAg nanoparticles, InP quantum dots, blue light leakage

## Abstract

In typical color-by-blue mode-based quantum dot (QD) display devices, only part of the blue excitation light is absorbed by QD emitters, thus it is accompanied by the leakage of blue light through the devices. To address this issue, we offer, for the first time, the applicability of AuAg alloy nanoparticles (NPs) as effective blue light absorbers in InP QD-based color-by-blue platforms. For this, high-quality fluorescent green and red InP QDs with a double shell scheme of ZnSe/ZnS were synthesized and embedded in a transparent polymer film. Separately, a series of Au/Ag ratio-varied AuAg NPs with tunable plasmonic absorption peaks were synthesized. Among them, AuAg NPs possessing the most appropriate absorption peak with respect to spectral overlap with blue emission are chosen for the subsequent preparation of AuAg NP polymeric films with varied NP concentrations. A stack of AuAg NP polymeric film on top of InP QD film is then placed remotely on a blue light-emitting diode, successfully resulting in systematically progressive suppression of blue light leakage with increasing AuAg NP concentration. Furthermore, the beneficial function of the AuAg NP polymeric overlayer in mitigating undesirable QD excitation upon exposure to ambient lights was further examined.

## 1. Introduction

Due to the strong merits of colloidal quantum dots (QDs), which include broad fine emission tunability, narrow (or color-pure) emissivity, outstanding photoluminescence quantum yield (PL QY), and solution-processibility, they have been regarded as the key visible emitters, particularly in next-generation display devices. While CdSe- and perovskite (e.g., CsPbX_3_, X = Cl, Br, I) based QDs possess exceptional bright and narrow emissivity attributes, their inclusion of toxic elements (i.e., Cd, Pb) arouses grave concerns given strict global environmental regulation standards [1,2,3,4]. In this regard, there was a growing consensus that InP QDs were unrivaled alternatives, till now, from the viewpoint of their environmental benignity and high PL performance. In particular, during the last decade, we have witnessed significant advances in the synthesis and core/shell heterostructural design of green- and red-emitting InP QDs [2,3,4,5,6,7,8]. The highest PL outcomes for both green and red InP QDs were most commonly achieved via a double shell scheme of ZnSe inner and ZnS outer shells, which resulted in >95% in PL QY and <37 nm in full-width-at-half-maximum (fwhm) [5,6,7,8]. Based on such progress, InP QDs were successfully utilized in both modes of color conversion (CC) and electroluminescence (EL). The CC mode-based display devices with relatively simple operating principles and structures were already industrialized by combining as an excitation source with blue light-emitting diode (LED) (often called QD enhancement film-based liquid crystal display (LCD)) or blue organic LED (OLED) (often called QD pixel-based OLED) while EL mode-based ones with rather complicated multilayered architectures along with exquisite processors are currently under active investigation aimed towards higher efficiency and longer lifetimes.

Metal nanoparticles (NPs) have been highlighted as an important class of nanoscale materials due to their intriguing optical, electronic, and catalytic properties. Among them, Au, Ag, and AuAg alloy NPs are the most studied since, not only can they be relatively facilely synthesized with well-defined morphology and colloidal form, in addition, they possess high chemical stability against oxidation and surface plasmon resonance (SPR) effects. Benefiting from such useful attributes, they have been actively explored in a variety of applications including sensing [9,10], catalysis [11,12], photovoltaic [13,14], and light-emitting devices [15,16,17,18,19]. The efficiency of light-emitting devices can be enhanced by taking advantage of localized SPR (LSPR), which is induced by the overlap between the LSP of metal NPs and the electromagnetic fields of the excitons of given emitters. Au and Ag NPs are most commonly utilized for this purpose and are integrated with green-emitting CdSe [15,16,17] and CsPbBr_3_ perovskite QD-LEDs (i.e., EL devices) [18,19]. Most recently, based on the composition-controlled AuAg NPs with their characteristic plasmonic absorption at the blue spectral range, the fabrication of LSPR-enhanced Cd-free blue-emitting ZnSeTe QD-LED was further demonstrated [20].

Meanwhile, in color-by-blue mode-based display devices, it is common for blue excitation light to not be completely absorbed by the green and red emitters accompanying the transmission (or leakage) of blue light through the devices. On that account, the use of color filters is necessary to cut off the blue component and thus only extract the desired green and red ones. Such blue leakage becomes more severe when InP QDs are used as emitters. In InP core/shell QDs, most of the blue light is absorbed by the InP core domain, whose sizes are ca. 2.0–2.3 nm for green-emissive cores, and ca. 3.3–3.5 nm for red-emissive cores [2,5,6,7,8]. Given that the light absorption coefficient scales with QD volume [21,22,23,24], substantial blue leakage naturally takes place in such tiny-sized InP QDs, although the smaller-sized green InP QDs suffer from more blue leakage than the large-sized red ones. In this contribution, we report an unprecedented applicability of AuAg alloy NPs as blue leakage suppressors in InP QD-based color-by-blue platforms. A series of AuAg alloy NPs with varied Au/Ag ratios are synthesized showing the composition-dependent systematic evolution of the LSP absorption peak. Among them, Au/Ag = 0.45-based AuAg NPs, whose absorption peak spectrally matches well with the blue spectral region, was chosen and embedded in a transparent polymeric matrix. These AuAg NP polymeric films with different AuAg NP concentrations are then integrated with green and red InP QD polymeric films. The resulting AuAg NP/InP QD film stack was placed remotely on a blue LED and the efficacy of the AuAg NPs in suppressing blue leakage was validated while varying the concentration of the AuAg NPs.

## 2. Materials and Methods

### 2.1. Synthesis of Green InP/ZnSe/ZnS QDs

In a synthesis of green-emissive InP cores, 0.15 mmol of indium acetate (In(OAc)_3_), 0.125 mmol of zinc acetate (Zn(OAc)_2_), and 0.7 mmol of palmitic acid (PA), along with 3.75 mL of 1-octadecene (ODE) were loaded in a three-neck flask. This mixture was degassed for 1 h while heating to 120 °C, and further heated to 240 °C under N_2_-purging. Then, 0.1 mmol of tris(trimethylsilyl)phosphine ((TMS)_3_P, 98%) in 1 mL of trioctylphosphine (TOP) was swiftly injected into the reactor and the core growth proceeded for 2 min at that temperature. After completing the core growth, the reactor was cooled down to room temperature and the resulting InP cores were precipitated by ethanol and re-dispersed in 1 mL of toluene. For the growth of the ZnSe/ZnS double shells, a Zn oleate solution (2.0 mmol of Zn(OAc)_2_ dissolved in 1 mL of oleic acid (OA), and 4 mL of ODE), 1.5 mmol of zinc chloride (ZnCl_2_), 3 mL of ODE, 3 mL of oleylamine (OLA), and 1 mL of InP core dispersion (in toluene) were placed in a three-neck flask. This mixture was degassed during heating to 120 °C for 30 min, and then its temperature was elevated to 260 °C under N_2_-purging. For the sequential growth of the ZnSe inner and ZnSe outer shells, a Se stock solution (1.5 mmol of selenium (Se) dissolved in 1.5 mL of TOP) was injected at that temperature, followed by the reaction at 260 °C for 30 min. Consecutively, ZnS outer shelling was implemented by introducing a Zn stock solution (0.7 mmol of Zn(OAc)_2_ dissolved in 0.425 mL of OA and 1.7 mL of ODE) and an S stock solution (0.5 mmol of sulfur (S) dissolved in 0.5 mL of TOP), followed by the reaction at 300 °C for 30 min. Subsequently, 0.8 mL of 1-octanethiol (OTT) was added into the reactor and the reaction was maintained at 220 °C for 30 min. Lastly, 1 mmol of Zn acetate dihydrate in 1 mL OA was continuously injected and reacted at 190 °C for 30 min. As-synthesized InP/ZnSe/ZnS core/shell QDs were repeatedly purified with a mixed solvent hexane/ethanol by centrifugation (9000 rpm, 10 min) and re-dispersed in hexane.

### 2.2. Synthesis of Red InP/ZnSe/ZnS QDs

In a typical growth of red-emissive InP cores, 1.2 mmol of In(OAc)_3_, 3.6 mmol of lauric acid (LA), and 20 mL of ODE were placed in a three-neck flask. This mixture was degassed for 1 h during heating to 140 °C, followed by cooling to room temperature under N_2_-purging. Then, 0.23 mmol of (TMS)_3_P dissolved in 7.77 mL of TOP was injected into the reactor, and the reaction proceeded at 300 °C for 1 h. To secure the desired sizes with red emissivity, the additional core growth was implemented consecutively by slowly co-injecting an In–Zn stock solution (3.6 mmol of In(OAc)_3_, 1.2 mmol of Zn(OAc)_2_ and 14.4 mmol of LA dissolved in 33 mL of ODE) and (TMS)_3_P stock solution (0.7 mL of (TMS)_3_P dissolved in 23.3 mL of TOP) at the rate of 16 mL/h and reacting at 320 °C for 2 h 45 min. After InP core growth was terminated, the reactor was cooled to room temperature and the resulting InP cores were precipitated by ethanol and re-dispersed in 10 mL of toluene. For the formation of the ZnSe/ZnS double shells, 5 mmol of Zn oleate solution (5 mmol of Zn(OAc)_2_ dissolved in 5 mL of OA and 4 mL of ODE), 2 mmol of ZnCl_2_, 4 mL of ODE, 6 mL of OLA, and 1 mL of purified InP cores in toluene were placed in a three-neck flask. After degassing the mixture during heating to 120 °C for 30 min, its temperature was raised to 320 °C under N_2_-purging. The ZnSe inner shelling was proceeded by an injection of a Se stock solution (3.5 mmol of Se dissolved in 3.5 mL of TOP) and reacting at 320 °C for 3 h. Consecutive ZnS outer shelling was then implemented by introducing a S stock solution (2.8 mmol of S dissolved in 1.4 mL of TOP) and reacting at 320 °C for 30 min. Subsequently, 0.5 mL of OTT was added into the reactor and the reaction was maintained at 220 °C for 30 min. Finally, 1.5 mmol of Zn acetate dihydrate in 1.5 mL OA was continuously injected and reacted at 190 °C for 30 min. As-synthesized red core/shell QDs were subjected to the same work-up process as for the above green ones.

### 2.3. Synthesis of AuAg NPs

A series of AuAg alloy NPs with different compositions were synthesized by varying the Au/Ag precursor feeding molar ratios of 0.45, 1, 2.3, and 3. First, 6 mL of ODE, 4 mL of OLA, and 1 mL of OA were placed into a three-neck flask and this mixture was degassed while heating to 120 °C for 30 min, followed by heating to 140 °C under N_2_-purging. For a typical synthesis of AuAg NPs with an Au/Ag ratio of 0.45, 0.22 mmol of silver acetate dissolved in 1.6 mL of ODE and 0.4 mL of OLA was rapidly injected into the above mixture and the reaction proceeded at 140 °C for 30 min. Subsequently, 0.1 mmol of gold chloride trihydrate dissolved in 1.6 mL of ODE and 0.4 mL of OLA was injected, followed by the reaction at that temperature for 30 min and a further 1 h 30 min at 250 °C. In the other cases, AuAg NPs with Au/Ag ratios of 1, 2.3, and 3, 0.16, 0.095, and 0.078 mmol of silver acetate and 0.16, 0.215, and 0.232 mmol of gold chloride trihydrate were used, respectively, while all other synthetic conditions remained unchanged. As-synthesized AuAg NPs were purified with excess ethanol by centrifugation (9000 rpm, 10 min) and re-dispersed in chlorobenzene.

### 2.4. Fabrication of InP QD-Polymer and AuAg NP-Polymer Films

For the preparation of ca. 500 μm-thick InP QD-polymer films, the optical densities of green and red InP/ZnSe/ZnS QDs dispersed in hexane were adjusted to be 1.0 at their own excitonic absorption peaks (i.e., 505 nm for green and 602 nm for red) and 1 mL of InP QD solution was homogeneously blended with 1 g of poly(dimethylsiloxane) (PDMS) and 0.3 g of silicon elastomer curing agent (SYLGARD 184, Dow Chem., Midrand, MI, USA). Following this, 1 mL of the above QD polymeric mixture was poured into an aluminum dish and slowly dried at 80 °C for 1 h.

For the preparation of ca. 300 μm-thick AuAg NP-polymer films, 200 mg of poly(methyl methacrylate) (PMMA, *M*_W_ = ~120,000) dissolved in 1 mL of chlorobenzene and 0.1 mL of AuAg NPs with concentrations of 1, 3 and 8 mg/mL in chlorobenzene were blended at room temperature for 20 min. The resulting homogeneous AuAg NP-polymeric mixture was poured into an aluminum dish and slowly dried at 80 °C for 1 h 30 min.

### 2.5. Characterization

The absorption spectra of InP QDs and AuAg NPs were measured by UV-visible spectroscopy (Shimadzu, UV-2450, Kyoto, Japan). The PL and PL excitation (PLE) spectra of the QDs were recorded with a 500 W Xe lamp-equipped spectrophotometer (PSI Co. Ltd., Darsa Pro-5200, Suwon-si, Republic of Korea). The PL QY of QDs was estimated in an integrating hemisphere with an absolute PL QY measurement system (QE-2000, Otsuka). High-resolution transmission electron microscopic (TEM) images of the InP QDs and AuAg NPs were collected by a JEM-2100F (JEOL Ltd., Akishima, Japan). The actual chemical compositions of the AuAg NPs were measured using a 15 kV-operating scanning electron microscope (SEM, JEOL-7800F) equipped with an energy-dispersive spectroscopy (EDS) system. The basic crystalline structures of green and red InP QDs along with a series of AuAg NPs were analyzed by X-ray diffraction (XRD, Rigaku, Ultima IV). Electroluminescent (EL) spectra of InP QD-polymer plates without and with a AuAg NP-polymer plate were acquired in an integrating sphere with a diode array rapid analyzer system (PSI Co. Ltd.).

## 3. Results and Discussion

For the subsequent color conversion experiments, we synthesized green and red InP QDs equipped with a well-known double shell scheme of ZnSe inner and ZnS outer shell. Figure 1a,b presents the normalized absorption (at excitonic absorption peak) and PL spectra of the green and red InP/ZnSe/ZnS QDs, respectively. The PL QY and fwhm values were 92 and 90% and 36 and 38 nm for green (530 nm) and red (620 nm) InP QDs, respectively, being in close proximity to those from state-of-the-art InP QDs in literature [5,6,7]. As shown in Figure 1c, the gap in absorbance at 450 nm between the two colored QDs was appreciable. Taking into account the volume scaling of light absorption coefficient [21,22,23,24], this is definitely associated with non-marginal differences in core size between the green (avg. 2.0 nm, Figure 1d) and red cores (avg. 3.5 nm, Figure 1f). In addition, note that TEM-derived average sizes of double-shelled green and red InP QDs were 7.2 (Figure 1e) and 8.9 nm (Figure 1g), respectively (also refer to Appendix A for their higher-magnification TEM images and Appendix A for their XRD patterns). It is worth mentioning that absorption spectral positions at 450 nm are different between the green and red InP core/shell QDs, as marked by a blue dotted line in Figure 1a. Unlike red QDs, the absorption at 450 nm for green QDs is positioned in close proximity to the absorption valley. Such dissimilar absorption spectral positions, in addition to different core sizes, jointly lead to a substantial gap in absorbance at 450 nm between two comparative InP QDs. To prepare free-standing QD composite films, green and red InP core/shell QDs were then individually embedded in a transparent polymer of PDMS (Figure 1h,i).

To tune the plasmonic absorption peak, the composition of the AuAg alloy NPs was varied by simply changing the Au/Ag precursor ratio in synthesis. TEM images of AuAg NPs upon applying Au/Ag ratios of 0.45, 1, 2.3, and 3 are shown in Figure 2a–d, respectively (also refer to Appendix A for higher-magnification images). Overall, their average sizes were similar in the range of 9.2–9.5 nm regardless of the Au/Ag ratio, although an increasing Au/Ag ratio led to a somewhat broader size distribution. The chemical compositions analyzed by an EDS clearly showed the steady reduction of Ag (L_α_) signals relative to the Au (M_α_) one with a higher Au/Ag ratio (Figure 2e) (which is also in line with the slight but systematic shift of reflection peaks in the XRD results in Appendix A), indicative of the successful composition control of AuAg NPs as intended. As a consequence, the plasmonic absorption peak of the AuAg NPs was systematically shifted to a longer wavelength as the Au/Ag ratio increased, specifically from 439 nm for Au/Ag = 0.45 up to 507 nm for Au/Ag = 3 (Figure 2f). It is also worth noting that among a series of AuAg NPs, the Ag-richest (i.e., Au/Ag = 0.45) ones exhibited the narrowest absorption band, which likely correlated with their highest degree of size monodispersity [25] (as mentioned earlier in the TEM results).

Among the above AuAg NPs with different Au/Ag ratios, the Au/Ag = 0.45-based AuAg samples that possessed the most appropriate absorption peak with respect to spectral overlap with the blue emission (e.g., 450 nm in wavelength) were chosen for the follow-up preparation of polymeric composite film of AuAg NP-PMMA. The concentration of AuAg NPs in the polymeric film was varied by preparing 1, 3, and 8 mg/mL of AuAg NPs in chlorobenzene and blending them with 200 mg/mL of PMMA chlorobenzene solution with an appropriate volume ratio (see Section 2). Plasmonic absorption peaks of all polymeric films of AuAg NPs were identical to that (439 nm) of their solution (Figure 3a). Meanwhile, the absorption tails appeared in AuAg NP-PMMA films, which is likely due to the absorption of the pure PMMA matrix (the inset of Figure 3a) and the scattering effects associated with possible inter-AuAg NP agglomeration in the matrix [26,27]. These AuAg NP polymeric films were then placed remotely on a 5 mA-driving blue LED, resulting in the successful suppression of blue emission leakage benefiting from the efficient blue absorption capability of AuAg NPs (Figure 3b). The suppression of blue leakage was also highly controllable by altering the concentration of AuAg NPs in the polymeric film, showing 54% for 1 mg/mL down to 2% for 8 mg/mL of AuAg NP concentration in leaked blue intensity relative to that of bare blue LED (Figure 3c).

To attest to the efficacy of AuAg NPs in the suppression of blue light leakage in color-by-blue mode, AuAg NP-PMMA film was integrated on top of the respective green and red InP QD-PDMS films, and this film stack was then placed remotely on a 5 mA-driving blue LED (Figure 4a). In the cases where the green and red QD polymeric films were without AuAg NP-PMMA film atop, substantial leakage of blue emissions was accompanied, while the leakage was more severe from the former (top panel of Figure 4b) than the latter film (top panel of Figure 4c). This disparity originates from the combined effects of nonmarginal core size difference along with the absorption spectral position at 450 nm between green and red InP QDs (as addressed earlier in Figure 1c). The superior capability of red to green InP QDs in blue light absorption naturally led to more intense red emissions than green ones. Emission peaks from green and red QD films were 540 and 631 nm, respectively, which were red-shifted from their solution emission peaks (i.e., 530 nm for green and 620 nm for red QDs in Figure 1b), which is attributed to the inter-QD light reabsorption in the QD films with relatively high QD concentrations [28,29,30]. As AuAg NP-PMMA film was integrated on top of green and red QD-PDMS films, leaked blue intensity was progressively abated with increasing AuAg NP concentration and became negligible for 8 mg/mL of AuAg NP concentration, being consistent with the results of Figure 3b,c. Also note that increasing AuAg NP concentration entailed a gradual decrease of color-converted green and red emissions, specifically upon applying 8 mg/mL of AuAg NP concentration showing the integrated spectral reductions of 45% for green and 38% for red InP QD emissions compared to the cases without AuAg NP polymeric film. This side effect is associated primarily with the presence of an absorption tail which extended over the longer wavelengths relative to the plasmonic absorption peak (Figure 3a). Somewhat less emission loss of red relative to green color is also understandable from the spectral attenuation of an absorption tail with a longer wavelength. As a supplementary experiment, commercial LCD green and red color filters (refer to Appendix A for their typical transmittance spectra) were employed instead of AuAg NP polymeric film for the same purpose under identical conditions. When compared, as in the insets of Figure 4b,c, 8 mg/mL-based AuAg NP polymeric film was found to be superior in the suppression of blue light leakage to LCD color filters for both green and red color-by-blue tests.

In the color-by-blue mode-based display device where QD emitters are placed in the front panel, QDs can be undesirably excited upon exposure to ambient lights [31,32]. This inevitably deteriorates the color purity and contrast ratio of the display device. In such a circumstance, the presence of AuAg NP polymeric overlayer can mitigate unwanted excitation of QDs by absorbing part of ambient lights (Figure 5a). To verify this, PLE spectra of green QD-PDMS films without versus with AuAg NP-PMMA overlayers having different AuAg NP concentrations were recorded with excitation lights directed on AuAg NP film side (Figure 5b). Green QD film only suffered from the excitation in a wide range of wavelengths. On the other hand, green QD film/AuAg NP film stacks exhibited an effective reduction of PLE intensity in a large spectral territory of QD excitation wavelengths (particularly corresponding to the absorption band of AuAg NPs), which was also proportional to the AuAg NP concentration. Figure 5c shows PL spectra of the same set of samples in Figure 5b, measured with an excitation of 450 nm directed onto the AuAg NP film side. The sharp emission band at 450 nm corresponds to the reflected excitation source, and the green QD emission band at 540 nm results from an excitation of 450 nm. Reflected excitation intensity became progressively attenuated with the increasing concentration of AuAg NPs acting as efficient blue absorbers, by which the blue photons reaching the QD film beneath lessened accordingly. As a result, upon applying an 8 mg/mL concentration of AuAg NP, green QD emission became eventually negligible. Furthermore, the same measurement as in Figure 5c was carried out for red QD film/AuAg NP film stacks, showing an identical trend in AuAg NP concentration-dependent spectral evolutions of reflected blue excitation and QD emissions (Figure 5d).

## 4. Conclusions

High-quality double-shelled green (530 nm) and red (620 nm) InP/ZnSe/ZnS QDs possessing 90–92% in PL QY and 36–38 nm in fwhm were first synthesized and embedded in a transparent polymer of PDMS. The composition-varied AuAg alloy NPs showing tunable plasmonic absorption peaks in the range of 439–507 nm were synthesized. Given an optimal spectral matching between the absorption of AuAg NPs and blue excitation light, Au/Ag = 0.45-based AuAg NPs were chosen for the preparation of PMMA-based polymeric composite films while varying the concentration of AuAg NPs. InP QD-PDMS film/AuAg NP-PMMA film stack was tested on a 5 mA-driving blue LED in a remote fashion. The results showed that the presence of a AuAg NP overlayer enabled the effective suppression of blue light leakage in both green and red CC mode tests, whose degree was proportional to the AuAg NP concentration. When the highest concentration of AuAg NPs (i.e., 8 mg/mL in this work) was applied, near-complete suppression of blue light leakage was possible. Furthermore, the utility of AuAg NPs as efficient light absorbers that can prevent unwanted QD excitation by ambient lights was validated.

## Figures and Tables

**Figure 1 materials-15-08455-f001:**
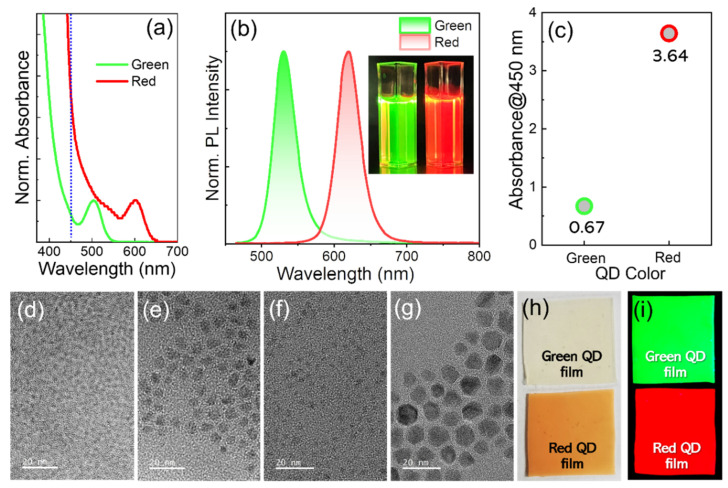
Normalized (**a**) absorption and (**b**) PL spectra of green and red InP/ZnSe/ZnS QDs. (**c**) Their absorbance values at 450 nm obtained from (**a**). TEM images of green-emissive (**d**) InP cores, (**e**) InP/ZnSe/ZnS core/shell QDs and red-emissive (**f**) InP cores and (**g**) InP/ZnSe/ZnS core/shell QDs (scale bars are 20 nm for all). Green and red QD polymeric films under (**h**) room light and (**i**) UV irradiation.

**Figure 2 materials-15-08455-f002:**
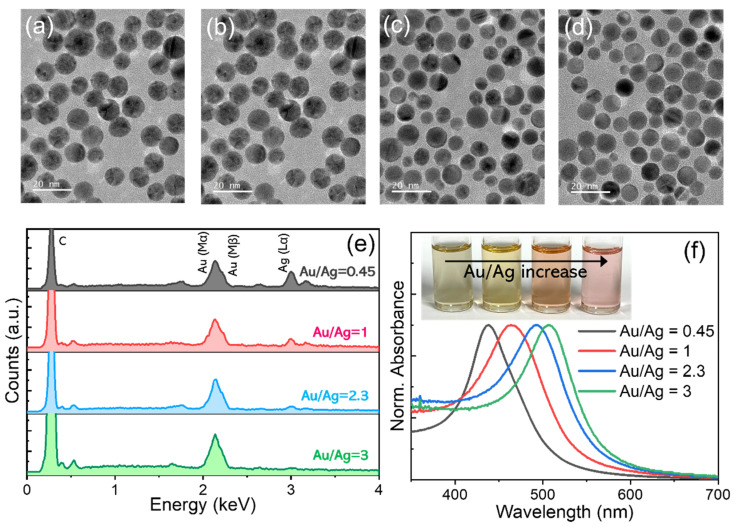
TEM images of AuAg alloy NPs synthesized with Au/Ag precursor ratios of (**a**) 0.45, (**b**) 1, (**c**) 2.3, and (**d**) 3 (scale bars are 20 nm for all). Comparison of (**e**) EDS spectra and (**f**) normalized absorption spectra (measured from dispersion in chlorobenzene) of a series of the composition-varied AuAg NPs.

**Figure 3 materials-15-08455-f003:**
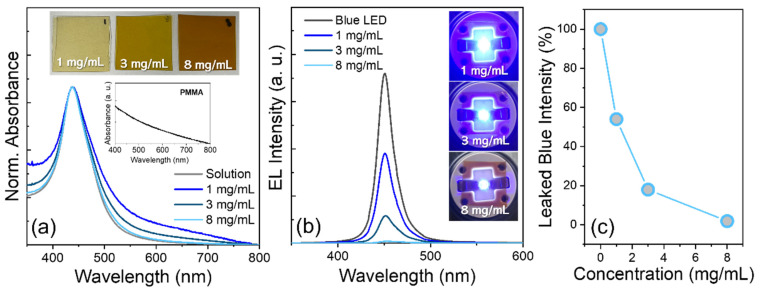
Comparison of (**a**) normalized absorption spectra of Au/Ag = 0.45-based AuAg NPs in solution (dispersion in chlorobenzene) and PMMA matrix film with different concentrations of AuAg NPs. The absorption spectrum of pure PMMA film is included in the inset. (**b**) EL spectra of 5 mA-driving blue LEDs without and with AuAg NP-PMMA films having different AuAg NP concentrations and (**c**) the corresponding leaked blue light intensity as a function of AuAg NP concentration.

**Figure 4 materials-15-08455-f004:**
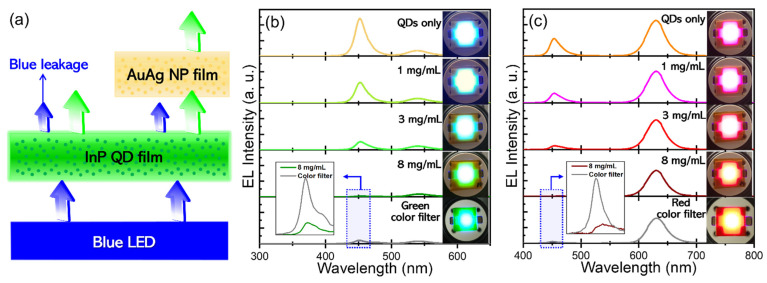
(**a**) Schematic illustrating the effective suppression of blue light leakage by the presence of AuAg NP polymeric film atop, (**b**) EL spectra and corresponding EL images of green, and (**c**) red QD-PDMS films integrated with AuAg NP-PMMA overlayers with different AuAg NP concentrations on a 5 mA-driving blue LED. Degrees of blue light leakage upon applying AuAg-PMMA film with the highest AuAg NP concentration (8 mg/mL) versus conventional LCD (**b**) green, and (**c**) red color filters were also compared (refer to the magnified plots of the insets of (**b**,**c**), corresponding to blue-dotted rectangles).

**Figure 5 materials-15-08455-f005:**
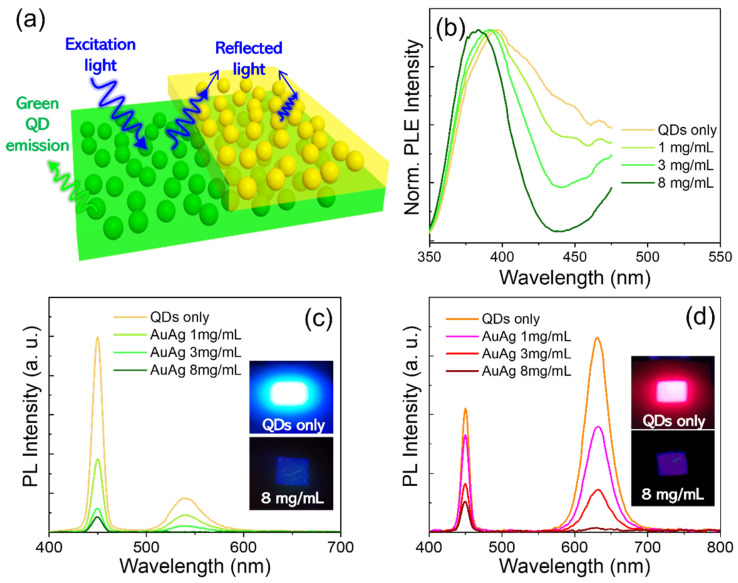
(**a**) Schematic showing AuAg NP polymeric film blocking ambient excitation lights, thus preventing undesired QD excitation. (**b**) Normalized PLE spectra of green QD-PDMS film/AuAg NP-PMMA film stack with different AuAg NP concentrations, measured with excitation lights directed onto the AuAg NP film side. PL spectra of (**c**) green and (**d**) red QD-PDMS film/AuAg NP-PMMA film stack having different AuAg NP concentrations, measured with an excitation of 450 nm directed on AuAg NP film side.

## Data Availability

Not applicable.

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
