# Peer review of "Effective Blue Light-Absorbing AuAg Nanoparticles in InP Quantum Dots-Based Color Conversion"

_materials, 2022, doi:10.3390/ma15238455_

Round 1
Reviewer 1 Report
In this paper, the authors use AuAg alloy nanoparticles as an effective blue light absorber in the InP QD blue light platform to eliminate blue light leakage. Considering the best spectral match between the absorption of AuAg NP and the blue excitation light, AuAg NPs based on Au/Ag=0.45 were selected to prepare PMMA based polymer composite films. The effectiveness of AuAg NP as an effective light absorber has been verified, which can prevent unnecessary QD excitation caused by ambient light. This work is very complete and has application prospects. However, I would recommend the authors to consider following suggestions.
1. As described in the manuscript, blue light is absorbed by noble metal NPs via localized surface plasmon resonance. It is known that the noble metal NPs absorb light to form localized surface plasmons and is thermalized by interface scattering and Landau damping. I want to know if the thermal effect of the Au/Ag alloy nanoparticles is significant and will affect the performance or lifetime of the devices?
2. Please give the OD value or transmittance value of the green and red color filters used in the experiments. By optical coating, the OD value of the interference type long-pass filter can reach 7 or even higher. I wonder what other advantages the Au/Ag nanoparticle films have in filtering blue light?
Reviewer 2 Report
The manuscript by Yeo et al reports the experimental work on the preparation of Ag/Au nanoparticles and their inclusion on green and red InP Quantum Dot (QD) devices in order to suppress the blue light emission which is usually leaked in this particular QD device. The green and red InP QDs (where it is double-shelled by ZnSe/ZnS) and Ag/Au were prepared using wet chemistry. The absorption and emission of their samples were taken using UV-Vis and PL/PLE spectroscopy, respectively. The authors claimed that the inclusion of Ag/Au nanoparticles can effectively suppress the blue emission from their QD device. The manuscript is well written and is of interest to the readers of materials MDPI’s journal, however, I have the following minor comments regarding their manuscript:
1. In Figure 3, page 6: The EL intensity of blue leakage LED is claimed to be reduced as the concentration of Ag/Au (ratio of 0.45) increased. This claim is based on the unnormalized intensity (see Fig 3b). How did the authors know that the intensity is indeed decreased intrinsically?
2. In Figure 4, page 7: The PL intensity of blue leakage LED that passes through the InP QD is claimed to be reduced as the concentration of Ag/Au (ratio of 0.45) increases. This claim is based on the unnormalized intensity (see Fig 4b and 4c). How did the authors know that the intensity is indeed decreased intrinsically? Why did the authors not try to plot the ratio between the peak at red/green emission and blue emission?
The manuscript could be published after these concerns are answered.

Reviewer 3 Report
This paper demonstrated the use of AuAg nano clusters for blue color management in displays. They show that with a proper Au/Ag ratio, the plasmonic absorption peaks can be well tuned to the emission wavelength of blue LEDs. The AuAg composite films show good performance in blue leakage and ambient light reflection reduction. Overall, this is a nice work that combined plasmonic effect with display applications. I would suggest minor revision. Followings are detailed comments.
1. AuAg is good for red emission as it absorbs less red light. However, from the absorption spectrum, there is still substantial green absorption, which though suppresses blue leakage but also wastes some of the green light. Is there a possible solution for this?
2. Earlier studies of plasmonic metal clusters for color filter application should be included in the introduction part.
3. Some of metal nano clusters show photoluminescence. How about those AuAg clusters?
4. Detailed HETEM and XRD analysis of the InP and AuAg nano particles should be provided.
